# Multivariate Framework of Metabolism in Advanced Prostate Cancer Using Whole Abdominal and Pelvic Hyperpolarized ^13^C MRI—A Correlative Study with Clinical Outcomes

**DOI:** 10.3390/cancers17132211

**Published:** 2025-07-01

**Authors:** Hsin-Yu Chen, Ivan de Kouchkovsky, Robert A. Bok, Michael A. Ohliger, Zhen J. Wang, Daniel Gebrezgiabhier, Tanner Nickles, Lucas Carvajal, Jeremy W. Gordon, Peder E. Z. Larson, John Kurhanewicz, Rahul Aggarwal, Daniel B. Vigneron

**Affiliations:** 1Department of Radiology and Biomedical Imaging, University of California, San Francisco, CA 94158, USA; robert.bok@ucsf.edu (R.A.B.); michael.ohliger@ucsf.edu (M.A.O.); zhen.wang@ucsf.edu (Z.J.W.); daniel.gebrezgiabhier@ucsf.edu (D.G.); tanner.nickles@ucsf.edu (T.N.); lucas.carvajal@ucsf.edu (L.C.); jeremy.gordon@ucsf.edu (J.W.G.); peder.larson@ucsf.edu (P.E.Z.L.); john.kurhanewicz@ucsf.edu (J.K.); dan.vigneron@ucsf.edu (D.B.V.); 2Helen Diller Family Comprehensive Cancer Center, University of California, San Francisco, CA 94158, USA; ivan.dekouchkovsky@ucsf.edu (I.d.K.); rahul.aggarwal@ucsf.edu (R.A.)

**Keywords:** hyperpolarized ^13^C MRI, pyruvate, advanced prostate cancer, metabolism, imaging features, overall survival

## Abstract

A novel hyperpolarized (HP) ^13^C MRI-specific multivariate framework was developed and evaluated in 16 patients with advanced or metastatic prostate cancer using radiomic-compliant feature definitions. From three basic metabolic maps (k_PL_, pyruvate summed-over-time, and mean pyruvate time), 316 features were extracted to construct univariate and multivariate risk classifiers. In the univariate analysis, patient subgroup with lower median k_PL_ had significantly longer PFS (11.2 vs. 0.5 months, *p* < 0.01) and OS (NR vs. 18.4 months, *p* < 0.05). A multivariate classifier also identified a lower-risk group with better outcomes. Despite limitations like small sample size and retrospective design, findings strongly support future investigation into prognostic values of these HP multivariate markers in prostate cancer.

## 1. Introduction

Recent years have seen a rapidly shifting landscape in the clinical management of advanced prostate cancer. Triplet regimens combining androgen receptor (AR)-targeted and cytotoxic drugs have prolonged survival in metastatic hormone-sensitive disease [1]; the approval of PARP inhibitors benefits select men carrying genetic vulnerabilities associated with DNA double-strand damage repairs [2], and PSMA-targeted Lutetium-177 radioligand therapy demonstrated an improvement in overall survival in a recent phase 3 trial in the post-taxane metastatic castration resistant (CRPC) setting [3].

These latest advances present oncologists and patients with a widening array of therapies both in the frontline and salvage settings. Concurrently, optimizing the selection, sequencing, and combination of treatments becomes more critical than ever. Non-invasive molecular imaging biomarkers offer an appealing methodology for the characterization of cancer biology, early assessment of therapeutic efficacy, and prognostic and predictive utility.

Hyperpolarized (HP) ^13^C-pyruvate MRI is a rapid 5 min, non-radioactive molecular MRI technology that has demonstrated both excellent safety and reproducible findings of cellular metabolism at over 15 research facilities globally in 1000+ cancer patients and normal volunteers [4,5]. The cancer Warburg effect—greatly increased lactate dehydrogenase (LDH)-driven malignant metabolism, quantifiable with HP ^13^C pyruvate-to-lactate conversion, is a hallmark of aggressive cancer biology, and is sensitive to early treatment response and resistance [6,7].

Despite the growing application of HP ^13^C MRI to characterize human prostate cancer [8,9,10,11,12,13,14,15], most of the existing analytics rely on the basic spatial distribution of pyruvate-to-lactate conversion rate (k_PL_) values [16,17], and currently, a paucity of effort is being made to systematically and quantitatively extract more complex underlying metabolic features—whether visible or invisible—such as shape, size, volume, texture, histogram statistics, and more.

Radiomic-style quantitative methods have become increasingly popular for extracting hundreds of rigorously defined and analytically calculated features from medical images [18,19]. These include simple first-order features describing voxel intensities such as mean, median, standard deviation, and entropy, second-order features portraying shape, texture, and patterns, as well as higher-order features derived by applying digital filters, time-frequency analysis or mathematical transforms to the original imaging data. Unlike artificial intelligence-based algorithms, which often operate through complex and opaque logic, radiomic-based feature extraction offers unambiguous definitions of HP MRI features that unifies interpretation and analysis. This provides a more direct understanding of their underlying molecular pathophysiological machinery, like Warburg metabolism, tumor microenvironment, tracer perfusion, and cellular uptake. There are suggestions that radiomics may be more robust for small-sample datasets and offer potential advantages in data-scarce scenarios compared with completely data-driven AI that directly learns obscure features from the native imaging data but requires a larger training set. This may make radiomics particularly appealing for many ongoing HP ^13^C studies, which are typically single-center or small multisite trials involving a relatively small number of subjects [20,21]. Nevertheless, a direct comparison is beyond the scope of our research.

Radiomic-style approaches have been utilized for feature extraction in conventional morphological MRI in different contexts such as breast cancer diagnosis and classification of brain tumor subtypes [22,23], as well as in a few molecular imaging modalities, including PET and SPECT [24,25,26,27]. Despite the smaller sample size available among the rapidly emerging field of HP ^13^C MRI research, there is growing enthusiasm for the use of such methodologies to conduct advanced image analytics just like its conventional and nuclear imaging counterparts, yet this angle remains largely unexplored [28]. To our knowledge, no prior work applied radiomic-style feature extraction for HP ^13^C MRI data.

This study evaluated a new radiomic-compliant multivariate quantitative framework to extract and characterize multiparametric features of metabolism (MFM) in advanced prostate cancer using cutting-edge abdomen/pelvis HP ^13^C MRI techniques and examined the correlation between these MFMs and clinical outcome measures using (1) a simple univariate and (2) a hypothesis-generating multivariate correlative analyses in a small set of patients.

## 2. Materials and Methods

### 2.1. Patient Demographics

Retrospective analysis was conducted on patients with locally advanced or metastatic prostate cancer prospectively enrolled in a HP ^13^C MRI study (NCT04346225) between November 2020 and May 2023. Key inclusion criteria included histologic and radiographic evidence of locally advanced or metastatic prostate cancer, presence of at least one target lesion amenable to HP ^13^C MRI, which met the conditions of either osseous lesion(s) visible on CT or MR or soft tissue lesion(s) measuring at least 1cm on cross-sectional imaging, and an ECOG status of 0 or 1. All participants (N = 16) were examined for and confirmed eligibility prior to enrollment. Written informed consent was obtained from all study participants. Follow-up data was available for all participants until either the event-of-interest, last day of follow-up, or lost to follow-up.

### 2.2. Hyperpolarization Methods

A mixture of 1.46 g of GMP-grade [1-^13^C]pyruvic acid (ISOTEC, MilliporeSigma, Burlington, MA, USA) and electroparamagnetic agent (EPA) AH111501 was loaded into a pharmacy kit and hyperpolarized for 2.5–3 h at 0.8 K in a 5T clinical-research DNP polarizer (SPINlab, GE healthcare, Chicago, IL, USA). The pyruvate mixture was then rapidly dissoluted, neutralized with an NaOH buffer, and underwent terminal sterilization. Prior to the delivery of each pyruvate dose to the patients, a pharmacist released the drug upon ensuring all quality control (QC) parameters were compliant with published safety/sterility criteria [29]. A summary of some recorded QC parameters included pyruvate concentration = 243 ± 13 mM, polarization = 33.9 ± 5.4%, pH = 7.6 ± 0.3, EPA concentration = 1.4 ± 1.0 mM, dose temperature = 30.3 ± 2.7 °C. Averaged time from dissolution to injection was = 53 ± 8 s.

### 2.3. Hyperpolarized ^13^C MRI Exam of Abdomen/Pelvis—Planning and Execution

The most recent clinical restaging scans (including CT, MR, and bone scans, subject to availability) were reviewed by HP ^13^C MRI scientists (HYC, RAB) and experienced clinical-research radiologists (MAO, ZJW) subspecialized in genitourinary cancers. An imaging plan was devised by identifying the target lesions of interest and corresponding coil placement via the consensus among these researchers.

The hyperpolarized exam was conducted using a clinical 3T MRI scanner (MR750, GE Healthcare, IL, USA) equipped with multi-nuclear spectroscopy package. The HP ^13^C imaging used a pairing of vest-shaped flexible transmitter and an 8-channel flexible array receiver (QTAR, Clinical MR Solutions, Brookfield, WI, USA), as previously described [30], which provided an approximately 32 × 24 × 22cm coverage in (LR, AP, SI) dimensions, respectively (Figure 1). The ^1^H MRI used the body coil transmit and a 21-channel flexible array (“AIR” coils, GE Healthcare) receive. RF transmit power on the ^13^C band was calibrated using a dimethyl silicone phantom prior to each MRI exam, and the center frequency was dialed in using a predefined scale-down factor from ^1^H frequency during the exam [31].

A 2D multislice EPI sequence was prescribed over the active region of the ^13^C coils targeting abdomen/pelvis with either 1.5 or 2 cm in-plane, 2 cm through-plane, and 3 s temporal resolutions [32]. Matrix size in (LR, AP, SI) was 22 × 22 × 15 for 1.5cm or 16 × 16 × 15 for 2cm resolutions. Excitation flip angles for pyruvate and lactate were set to 15° and 30°, respectively. Alanine images (30° flip angle) were additionally acquired for liver metastasis cases but excluded from the formal analysis due to its undetermined biological role in prostate cancer [33].

Proton MRI protocol of the abdomen or pelvis was tailored to the targets of interest per patient, which entailed a pre-contrast and multiphase post-contrast T_1_ fast-spoiled-gradient-echo sequences, T_2_ single-shot FSE, diffusion weighted imaging with b = 50 and 800, and a three-point Dixon sequence providing fat, water, and B0 field maps. A T_2_-short-tau-inversion-recovery imaging sequence was used to detect bone marrow abnormalities. In cases involving liver metastases, hepatobiliary phase images were acquired 15–20 min after intravenous contrast (gadoxetic acid) when feasible.

### 2.4. Multiparametric Feature Extraction and Analysis

Tumor ROIs were segmented using 3D Slicer [34] semi-automatically by a researcher with 7+ years of experience interpretating both HP ^13^C and ^1^H MRI of advanced prostate cancer (HYC). The observer first reviewed the acquired ^1^H images alongside clinical MRI report and prior CT/MRI images to identify any existing and new cancerous lesions. An ROI is defined for any lesion found within the imaging volume, including metastatic lesions to the bone, visceral organs, lymph nodes, and primary tumor site when present. The ROIs were delineated with reference to a ^13^C/^1^H pseudocolor overlay heatmap and representative ^1^H anatomical series, displayed side by side; this was done primarily in a qualitative fashion based on lesion shape, size, and extent on underlying ^1^H images but was extended whenever hotspots with elevated k_PL_ exceeded the boundaries of ^1^H MRI lesion. Additionally, the identified lesions were annotated on ^1^H/^13^C images, summarized into a PowerPoint presentation for each patient, reviewed by board-certified and fellowship-trained abdominal radiologists (MAO, ZJW) as well as key research personnel (IdK, RAB, DBV), and discussed during monthly meetings. Ambiguous lesion findings on ^1^H MRI were consulted with the aforementioned radiologists for either confirmation or exclusion. Co-registration between ^1^H and ^13^C images were visually confirmed using key anatomical features such as the aorta, iliac arteries and kidneys.

PyRadiomics (v3.1.0) [35], an open-source Image Biomarker Standardisation Initiative (IBSI, a consensus effort to standardize the definition and extraction of imaging features aiming to improve reproducibility of radiomic research)-compliant software package based on Python (v3.8), was used to extract MRI features from three source HP parametric maps, namely pyruvate-to-lactate conversion rate (k_PL_), pyruvate signal summed-over-time (AUC), and mean pyruvate time (MT). To homogenize the data, these source maps were resampled from their native resolutions to 5 mm isotropic voxels. A total of 316 multiparametric features of metabolism (MFM) were extracted using PyRadiomics. Classes of extracted features included Pyradiomics’ default first-order features, as well as higher-order classes gray-level dependence matrix (GLDM), gray-level run length matrix (GLRLM), and gray-level size zone matrix (GLSZM).

k_PL_ was calculated using an inputless two-site exchange model [36]. Mean pyruvate time was defined as the center of mass of the time-resolved pyruvate pharmacokinetic signal curve. Gray scale discretization was implemented using empirically determined bin widths of 0.001, 0.02, and 0.1 for k_PL_, AUC, and MT, resulting in 43, 100, and 471 bins, respectively. PET literature suggested fixed bin width yielded more reproducible measures than fixed bin count [37].

### 2.5. Survival Analyses Using Uni- and Multivariate Models

Clinical information was extracted from the medical chart and included patient age, race and ethnicity, clinical TNM staging, hormonal status (hormone sensitive versus castration-resistant), serological biomarkers such as prostate-specific antigen (PSA), LDH, and alkaline phosphatase (ALP), and histological subtypes from any metastatic biopsy. Note: ALP readings were missing from two participants and replaced with cohort median for the statistical analysis.

Progression-free survival (PFS) was defined as the interval between the date of HP MRI and the date of either clinical or radiographic progression, or death. Overall survival (OS) was similarly defined as the interval between the date of HP MRI and the date of death. Patients who continued to be followed up and remained event-free were censored on the last day of follow-up (October 2023), while those lost to follow-up were censored at the last date they were either known to be alive or event-free. Progression was determined according to the judgment of the treating oncologist, which typically depended on one or more of the following factors: (1) radiographic progression, (2) worsening of symptoms, and (3) switching treatment or management.

Survival analyses using uni- and multivariate Cox proportional hazards models examined the prognostic values of these MFMs. In the univariate model, *p*-value and Harrell’s concordance index (C-index) of PFS and OS were calculated for each individual MFM and serological feature (PSA, LDH, ALP) as correlative metrics to clinical endpoints. Regression was conducted with Cox proportional hazards model provided in the MATLAB toolbox (R2016a) which performed the likelihood ratio test to estimate the β coefficient, *p*-values, and covariance matrix, taking the PFS, OS data, and censoring array as inputs. C-index is the equivalent of area under receiver operating curve for survival models dealing with censored data [38].

In the multivariate case, all 316 MFMs and 3 clinical serological features were initially incorporated. Feature selection (dimensional reduction, an essential process in multivariate models for identifying important MFMs and eliminating redundancy) was performed by ranking features’ importance by C-index, followed by agglomerative (hierarchical) clustering [39] classifiers to eliminate highly correlated (i.e., redundant) features.

In the ranking step, features with top 20% highest adjusted C-index were retained, resulting in 65 remaining features. The adjusted C-index was empirically defined as a weighted sum of 0.8 × C-index_{PFS}_ + 0.2 × C-index_{OS}_, based on the maturity of the PFS and OS data (maturity of each survival metric was qualitatively defined as a combination of the median follow-up time, number of events, and censoring rate). The clustering step used a distance metric, defined as *1 − absolute Pearson correlation*, to construct a hierarchical dendrogram “tree” with 4 branches and multiple leaf nodes on each branch, with the highest mutually dependent features located on the same leaf node [40]. The tree was reorganized into a correlogram consisting of 4 clusters for visualization, where one feature with the highest C-index was selected from each cluster. Finally, a metabolic prognostic score (MPS) was defined using the 4 selected variables and adjusted for patient age, as detailed in the results.

To determine the stability of the multivariate model and detect potential overfitting, a sensitivity analysis was conducted in a cross-validation manner. One patient dataset was withheld at a time, and Cox model regression was conducted on the remaining fifteen datasets to estimate the MPS risk classifier coefficients. Normalized standard deviation of each MPS coefficient were calculated across 16 iterations.

## 3. Results

### 3.1. Clinical Characteristics

Sixteen consecutive prostate cancer patients who underwent HP ^13^C MRI of the abdomen/pelvis were included in this analysis. At the time of scans, 11 patients had CRPC and 5 had hormone-sensitive prostate cancer (HSPC). Staging-wise, four patients with distant metastases had visceral metastases (M1c), eight had osseous disease with or without nodal metastases (M1b), and one had nodal metastases only (M1a). Of the locally advanced disease category, two had nearby organ invasion (T4), and one had seminal vesicle invasion (T3b) (Table 1). On a per lesion basis, the analysis included 90 metastatic lesions of the bones, 12 of the liver, 14 of the lymph nodes, and 2 of other soft tissues such as the rectal wall; there were also seven primary tumor lesions of the prostate. Of the six patients who underwent metastatic biopsy, exclusively adenocarcinoma was found in four, adenocarcinoma with focal neuroendocrine prostate cancer (NEPC) was found in one, and one was negative on histopathology (attributed to sampling error).

### 3.2. Univariate Model Detected Significant Correlation Between k_PL_ and Clinical Endpoints

The 316 MFMs extracted out of the three basic HP parametric maps contained first-order features such as mean, median, variance, kurtosis, and total metabolic volume (TMV, total tumor volume with measurable k_PL_ values), as well as higher-order features including GLDM, GLRLM, and GLSZM. With a median follow-up of 22.0 months, k_PL,median_ significantly correlated with both PFS (*p* < 0.01) and OS (*p* < 0.05). Corresponding Kaplan–Meier statistics revealed significantly longer median PFS (11.2 vs. 0.5 months, C-index = 0.813, hazard ratio [HR] = 5.1; 95% confidence interval [CI]: 1.3–20.3) and OS (NR vs. 18.4 months, C-index = 0.879, HR = 6.1; CI: 1.1–32.3) in the lower- than the higher-k_PL_ subgroup, dichotomized by a k_PL,median_ cutoff value of 17 (ks^−1^) (Figure 2).

On the other hand, there was no significant difference in k_PL,median_ as a function of hormonal status. The CRPC subgroup had k_PL,median_ = 15 ± 11 ks^−1^, and the HSPC subgroup had k_PL,median_ = 16 ± 6 ks^−1^ (*p* > 0.5).

Interestingly, serological markers, including PSA, LDH, and ALP, were not significantly associated with PFS or OS (*p* > 0.05) in this small cohort (Table 2).

Pearson correlative statistics (Figure 3 and a full coefficient table in Appendix A) identified a strongly negative correlation between k_PL_ kurtosis and OS (r = −0.75), and moderately negative correlations between k_PL,median_ with OS (r = −0.45), k_PL,max_ with both PFS (r = −0.48) and OS (r = −0.54), and kurtosis with PFS (r = −0.37). Correlative metrics between MFMs and serological markers revealed a moderate correlation between TMV and PSA (r = 0.55), and weak correlations between k_PL,median_ and both PSA (r = 0.08) and LDH (r = 0.08); similarly, weak correlations were found between k_PL,max_ and PSA (r = 0.12) and LDH (r = −0.09).

### 3.3. Multivariate Model Offered a Provisional Approach for Larger Future Datasets

The 65 retained MFMs (top 20% highest adjusted C-indices) had a median adjusted C-index of 0.77 (range: 0.74–0.85). Agglomerative clustering identified four clusters consisting of highly correlated, redundant features, and one representative feature were selected from each while eliminating the remainders (Figure 4). Three MFMs and one serological feature were selected, namely *kPL_median, pyrAUC_original_shape_Elongation, kPL_original_firstorder_90Percentile*, and *PSA*, respectively. *kPL_median* was the median k_PL_ across ROIs, *pyrAUC_original_shape_Elongation* represented square root aspect ratio derived from the lengths of the largest and second largest principal component axes extracted from the pyrAUC map, and *kPL_original_firstorder_90Percentile* stood for 90th percentile signal intensity from the k_PL_ map. These three MFMs all belonged to the first-order feature class. The multivariate Cox proportional hazards model was constructed using a composite risk score—metabolic prognostic score (MPS), defined as a weighted combination of these four variates using the b coefficients extracted from the model regression and adjusted for patient age. A lower MPS means more favorable prognosis.
***Metabolic Prognostic Score (MPS)*** = 0.269 × ***kPL_median*** +                               
−3.069 × ***pyrAUC_original_shape_Elongation*** +                               0.165 × ***kPL_original_firstorder_90Percentile*** +                0.0063 × ***PSA*** +
              −0.0874 × ***Age***

Similarly to the univariate approach, the age-adjusted MPS significantly correlated with both PFS (*p* < 0.002, C-index = 0.902) and OS (*p* < 0.05, C-index = 0.951), with longer PFS (NR vs. 2.4 months) and OS (NR vs. 18.4 months) in the lower vs. higher MPS categories, dichotomized by the cohort median MPS = −0.581 (Figure 5).

## 4. Discussion

For the first time, quantitative multivariate MRI features evaluating Warburg metabolism, pyruvate uptake, and bolus profiles were methodically extracted from whole-abdomen/pelvic HP ^13^C-pyruvate MRI of men with advanced prostate cancer. This new framework was enabled by a combination of commercially available ^13^C + ^1^H coil array hardware, a rapid volumetric EPI acquisition, and an open-source radiomic-compliant software package. The feasibility of HP ^13^C MRI as a safe, 5 min addition to standard staging/restaging ^1^H MRI scans and the accompanying capability to calculate hundreds of MFMs—from as simple as k_PL,median_, k_PL,max_ to composite, multivariate risk classifiers like MPS—may shed light on the complex metabolic behavior of aggressive prostate cancer variants associated with poor outcomes. The study results highlight HP MRI’s unique potential to provide highly prognostic relevant metabolic information noninvasively.

The early signal of statistical correlation between MFMs and clinical endpoints proved particularly encouraging, considering the substantial heterogeneity among this small study population—clinical stages ranging from locoregional to distant metastases; and variable scan timings corresponding with clinical status anywhere from remission to stable or progressive disease.

k_PL,median_ was experientially chosen as the risk classifier for the univariate model owing to its established role as a HP biomarker of Warburg metabolism. Similarly, k_PL,max_ turned out to be a statistically significant univariate predictor. However, we elected not to report an exhaustive predictive analysis searching through each of the 316 HP ^13^C multiparametric MRI features, out of the caution that many false positive findings can arise by sole probability without a strong supporting biological rationale. On the other hand, many MFMs calculated from the k_PL_ base map will conceivably turn out positive due to redundancy and yet may not provide much additional mechanistic insight beyond simple k_PL,median_. Such an analysis would be better suited for the presented multivariate survival model intended for hypothesis generation.

Another interesting observation was that k_PL_ value alone strongly discriminated patients with more favorable from unfavorable prognosis. By contrast, a recent work examining the prognostic value of PSMA-PET [41] found the SUV value to have a much weaker effect compared with other PSMA-derived metrics such as lesion count, tumor volume, and presence of distant metastases (see reference [41]’s Figure 1 forest plot and Figure 2 nomogram). Dysregulated glycolytic metabolism is essential for driving tumor cell survival, proliferation, and invasion, and k_PL_ is known to be highly associated with cancer aggressiveness and treatment resistance [4,5]. PSMA expression levels are often modulated by prolonged hormonal therapy and may not be a direct readout of cancer aggressiveness in heavily pretreated disease [42]. The growing use of radioligand therapies could further complicate PSMA interpretation.

A few key observations from the Pearson correlation matrix (Figure 3) are worth highlighting. The weak correlations between k_PL_-related MFMs (k_PL,median_, k_PL,max_) and serological markers (PSA, LDH) suggested that these MFMs provided independent prognostic information not accessible via conventional clinical markers. High k_PL,kurtosis_ reflected tumor metabolic heterogeneity resulting from treatment-induced clonal evolution and increased mutational burden [43]. This presumably gave rise to aggressive treatment-resistant variants associated with poor outcomes. The moderate correlation between metabolic tumor volume (MTV) and PSA (r = 0.55) possibly reflected these biomarkers’ role as estimators of tumor burden.

While k_PL,median_ and k_PL,max_ were significantly associated with both PFS and OS, the same association was not found in this study for established serological prognostic markers such as PSA, LDH, and ALP [44,45]. This could possibly suggest that select HP MFMs have a larger effect for detection in this relatively small cohort. Nevertheless, this preliminary observation needs to be validated in a larger population. Along the same line, whether these MFMs are stronger predictors and/or independent predictors of clinical TNM staging remains to be answered in the future.

Many aggressive variants of prostate cancer, such as AR-independent and NEPC subtypes, are often characterized by low serum PSA secretion but rapid progression and poor prognosis [46,47]. Although four out of five subjects in this study who underwent successful metastatic biopsy found only adenocarcinoma, these biopsies are not immune to sampling error—the presence of inter- and intratumoral heterogeneity is well understood [48]; mixed phenotypes are particularly common in heavily pretreated disease [47,49]. In addition, AR staining was not routinely performed as a part of our clinical practice. Furthermore, these biopsies were not routinely performed at the time of HP MRI. These uncertainties pose challenges to directly link MFM-PSA disassociation identified in this study to histological evidence of aggressive variants.

Although LDH is the key upregulated enzyme driving the cancer Warburg effect, elevated serum LDH can be linked to a range of non-specific biological processes such as inflammation and immune response [50,51]. HP MRI features could provide spatial specificity through tumor localization that circulating LDH markers do not. Also, as a biomarker of bone matrix breakdown, ALP level is more meaningful in bone-dominant or bone-only disease [52] and thus may not be as prognostic in our cohort of 16, which consisted of a heterogeneous mixture of bone, soft tissue, and visceral metastases.

Prostate cancer is a heterogeneous malignancy with complex underlying biology and diverse clinical presentation, histological subtypes, and gene expression [53]. Accordingly, most established clinical risk classifiers and nomograms are multivariate [54,55]—accounting for clinical, serological, and histological parameters. These strongly support the idea of a multivariate radiomics approach toward HP ^13^C MRI of advanced prostate cancer. Keeping in mind the small size of our pilot study, we observed that the multivariate MPS was dependent on k_PL_ levels and the tumor ROI’s shape elongation (aspect ratio). The shape factor could be related to tumor invasiveness. It was reassuring that PSA, an established clinical biomarker, was identified as one of the four classifier variables in the multivariate analysis, underscoring its role reflecting tumor burden despite the possible discordance commonly found in aggressive histological variants. Although MPS showed higher concordance (C-index) with outcomes over k_PL_ alone, the advantages of our multivariate model over its univariate counterpart requires an external validation dataset to be definitively established.

Cross-validation of the multivariate model showed varying stability of each MPS coefficient—normalized standard deviation ranged from 0.33 to 0.57 (Appendix A). This suggested that caution should be exercised about the limited sample size and associated possibility of false discovery. Although the numerical coefficients may suffer from a certain degree of overfitting, our primary intention is to propose and showcase the generalizable approach of whole abdomen/pelvis HP MRI, metabolic feature extraction, selection, model construction, and data representation and visualization that will be readily applicable to future collaborative, multicenter HP studies to establish more refined multivariate cancer metabolic risk classifiers.

Taken together, the value proposition for this multivariate approach is its generalizability toward larger studies. Additionally, our initial observations uncovered tantalizing leads to identifying potentially important risk variables predictive of clinical outcomes.

Our two-step feature selection process consisted of feature ranking followed by agglomerative clustering. This was chosen out of intuitiveness and the possibility of stepwise graphical visualization. More sophisticated methods for dimensional reduction, such as random forest classifier [56], LASSO regression [57], and recursive feature elimination [58] could offer better performance. Comparison of these dimensionality reduction techniques in conventional radiographs (e.g., LASSO, PCA, recursive feature elimination) showed similar performances and each useful for certain scenarios [59,60], but drawing such a conclusion specifically in the HP ^13^C MRI context requires more data than that available from our study.

The fast-growing field of deep learning may benefit complex HP ^13^C MRI feature extraction and classification tasks. However, a direct comparison between conventional radiomics and convolutional neural networks is outside of our scope and better suited for a larger future study.

A few limitations should be acknowledged for the current research, including limited sample size, referral bias and retrospective endpoint definitions. A major limitation lies in the small sample size, which precluded division of the 16 subjects into training and validation sets to rigorously evaluate the model performance and may limit the generalizability of our models. Nevertheless, these initial positive findings provide optimism that the necessary sample size for validation will be addressed in future multicenter collaborative HP research.

As this was a retrospective analysis, the scan timings and the corresponding clinical statuses were not prespecified and were indeed somewhat heterogeneous. Common fallacies of retrospective studies like ours may include lack of randomization, misclassification bias, missing data entries, confounders, and misidentification of cause–effect [61,62]. While OS is less ambiguous and minimally impacted by retrospective endpoint design, surrogate endpoints like PFS can be confounded by recall bias, informative censoring, and assessment time bias; these remained to be addressed in a prospective study.

Interobserver agreement was not assessed in this small-sample pilot research, which may limit ROI reproducibility. This was partially mitigated by a standardized approach for ROI identification and delineation in the single-observer scenario, and routine consensus imaging data and lesion review by radiologists and key researchers. The researchers were not blinded to the patients’ medical history or any prior (conventional) scan findings at the time of the HP ^13^C MRI. However, this impact is considered minimal due to the observational nature of this research.

The survival metrics calculated in this study reflect those with relatively good functional status (per inclusion criteria) and could be positively biased against the population median which would have included older, frailer and/or more symptomatic individuals. Care should be taken when comparing these PFS and OS findings to other observational studies or clinical trials.

Although proton MRI was performed alongside HP ^13^C MRI for all patients, the ^1^H MRI protocols varied depending on which body part was being targeted in an attempt to optimize our approach. Such variability precluded incorporation of proton-MRI features (based on T_1_-, T_2_-, and diffusion-weighted imaging) into this research. Establishing a generic ^1^H protocol similar to the published whole-body MRI for prostate cancer evaluation can potentially make future ^1^H scans more routine and uniform [63,64]. The combination of both HP ^13^C and standard ^1^H MRI features would be an interesting topic for future study.

## 5. Conclusions

For the first time, multiparametric features of metabolism (MFM) were systematically extracted from hyperpolarized (HP) ^13^C-pyruvate MRI of men with advanced prostate cancer, enabled by the latest advancements of HP ^13^C + ^1^H array hardware combinations and novel metabolic MRI approaches offering full abdominal or pelvic coverage. Preliminary analysis of a small set of patients found that select MFMs—including intratumoral HP ^13^C pyruvate-to-lactate conversation rate k_PL_—were significantly associated with clinically meaningful outcome measures. These encouraging data strongly support further pursuit of the prognostic values of HP ^13^C MRI-derived metabolic features in prospective studies.

## Figures and Tables

**Figure 1 cancers-17-02211-f001:**
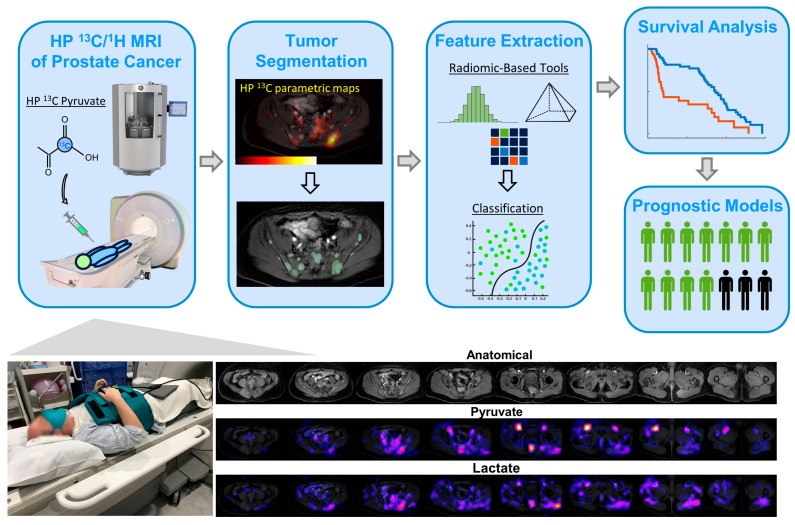
Whole abdominopelvic HP ^13^C metabolic MRI using cutting-edge hardware and techniques enabled extraction and characterization of radiomic-compliant multiparametric features of metabolism (MFM) in advanced prostate cancer.

**Figure 2 cancers-17-02211-f002:**
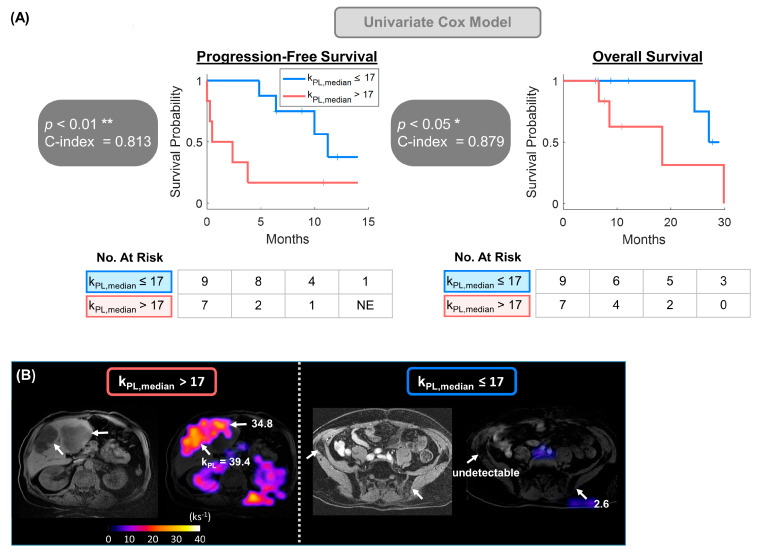
HP multiparametric features of metabolism (MFM) k_PL,median_ was selected and applied to univariate Cox proportional hazards model. (**A**) Kaplan–Meier analysis revealed substantially longer PFS and OS in patients with k_PL,median_ ≤ 17 ks^−1^ (median PFS: 11.2 vs. 0.5 months; median OS: NR vs. 18.4 months). (**B**) Example cases illustrating metastatic prostate cancer patients with k_PL,median_ higher and lower than the dichotomized cutoff value of 17. The patient on the left had high-volume liver metastases with k_PL,median_ = 24.8, whereas on the right had low or undetectable k_PL,median_ across the osseous pelvic metastases. Images shown were k_PL_ heatmaps overlaid on ^1^H T_1_-FSPGR references. * *p* < 0.05, ** *p* < 0.01.

**Figure 3 cancers-17-02211-f003:**
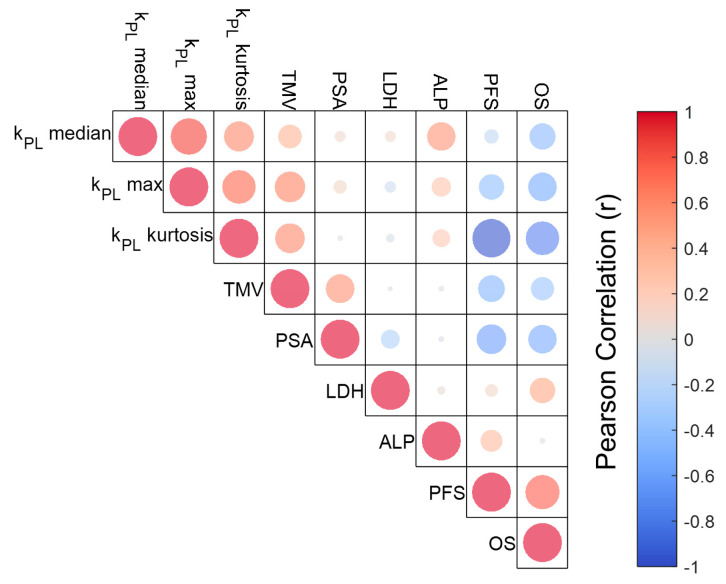
Pearson correlation matrix revealed a strongly negative correlation between k_PL_ kurtosis and OS, and moderately negative correlations between k_PL,median_ with OS, k_PL,max_ with both PFS and OS, and kurtosis with PFS. Red and blue colors on the heatmap represent positive and negative correlations, respectively; the circle size represents the correlation strength. TMV: Total metabolic volume. PFS: Progression-free survival. OS: Overall survival.

**Figure 4 cancers-17-02211-f004:**
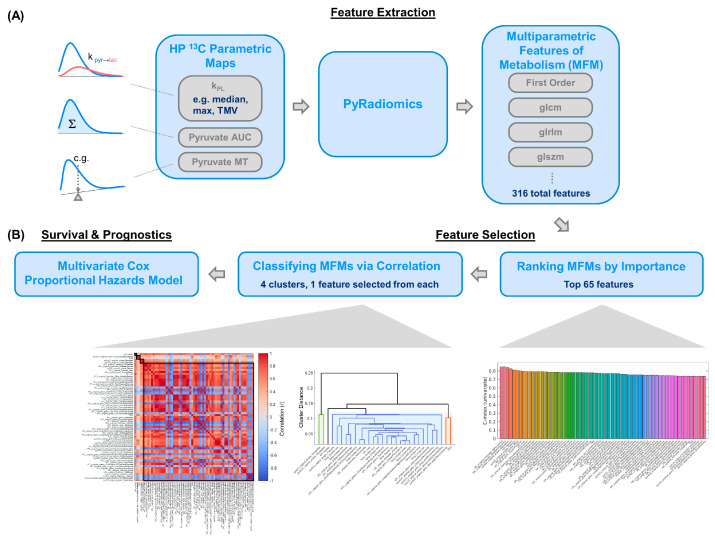
(**A**) Framework for extraction, selection, and classification of multiparametric features of metabolism (MFMs) consisted of a feature extraction step where open-source PyRadiomics package extracts MFMs from the three source HP parametric maps (k_PL_, pyruvate AUC and MT), followed by a feature selection step consisting of C-index ranking of MFM importance and an agglomerative clustering classifier. (**B**) Four clusters consisted of highly correlated, redundant features, and one representative feature was selected from each while eliminating the remainders.

**Figure 5 cancers-17-02211-f005:**
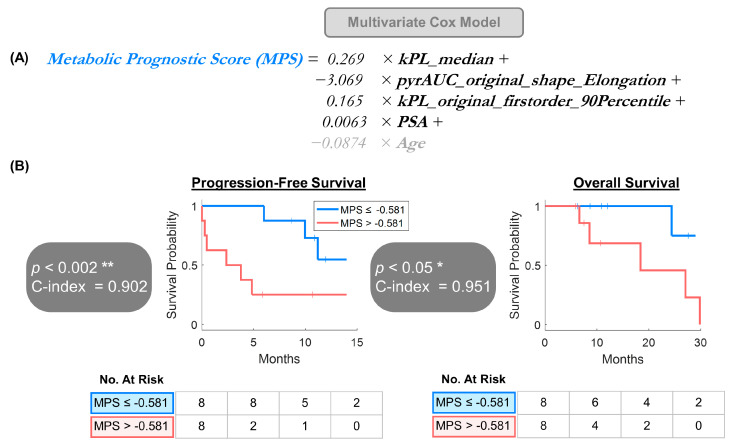
Hypothesis-generating multivariate survival analyses using (**A**) metabolic prognostic score (MPS) derived from multivariate Cox proportional hazards model, adjusted for patient age. (**B**) MPS was significantly associated with both PFS (*p* < 0.002) and OS (*p* < 0.05), with longer median PFS (NR vs. 2.4 months) and OS (NR vs. 18.4 months) for lower-MPS vs. higher-MPS patients, dichotomized by the cohort median MPS = −0.581. * *p* < 0.05, ** *p* < 0.01.

**Table 1 cancers-17-02211-t001:** Clinical characteristics of the study participants.

Characteristic	Value
Participants (N = 16)	
Age (years)	69 ± 10 (52–88)
Staging	
Local-regionally advanced	
T3	1
T4	2
Metastatic	
M1a	1
M1b	8
M1c	4
Hormonal Status	
Sensitive	5
Resistant	11
Laboratory Markers	
PSA	13.9 (0.05–336.0)
LDH	208 (139–422)
ALP	76 (37–190)

PSA: Prostate-specific antigen. LDH: Lactate dehydrogenase. ALP: Alkaline phosphatase.

**Table 2 cancers-17-02211-t002:** In the univariate analysis, k_PL_ max, median, and kurtosis were significantly associated with both PFS and OS, outperforming established clinical prognostic markers PSA and LDH. This preliminary finding suggested the HP ^13^C multiparametric features of metabolism (MFM) may have stronger effect for detection in this small dataset than these clinical markers.

Feature	Likelihood *p*-Values	
	PFS	OS
k_PL_ max	* 0.024	* 0.031
k_PL_ median	** 0.008	* 0.048
kurtosis	* 0.023	* 0.029
TMV	0.106	* 0.028
PSA	0.194	0.412
LDH	0.601	0.342
ALP	0.853	0.821

* *p* < 0.05, ** *p* < 0.01.

## Data Availability

Deidentified data can be made available upon request made to the corresponding author.

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
