# Peer review of "Multivariate Framework of Metabolism in Advanced Prostate Cancer Using Whole Abdominal and Pelvic Hyperpolarized 13C MRI—A Correlative Study with Clinical Outcomes"

_cancers, 2025, doi:10.3390/cancers17132211_

Round 1
Reviewer 1 Report
Comments and Suggestions for Authors
The authors described a framework to evaluate the association between multi-parametric features of metabolism using radionics-based approach. This was in prostate cancer patients undergoing hyperpolarized 13C pyruvate MRI.
This work is novel and it's the first work to describe this approach where multiple metabolic parameters can be extracted from 13C pyruvate readouts and then analysed using radiomics. This work shows that beyond pyruvate to lactate conversion metrics, more intuitive information about the disease characteristics could be extracted and potentially guide disease-making. This makes it impactful work.
Few things:
Is this work post-hoc analyses or retrospective study. It's probably just semantics. Was it a pre-planned, post-hoc study ?
While radiomics MFM based analyses is interesting and in this case useful, I'm sure the authors are aware about the pitfalls of radiomics. In fact, some of these were described in their discussion. Would deep learning (for the imaging data) or machine learning tools be more useful ? Though this reviewer is aware of the modest sample size and this might leading to overfitting amongst other issues.
Otherwise, this work is well-written, technically quite robust and informative.
Reviewer 2 Report
Comments and Suggestions for Authors
The abstract mentions promising results correlating imaging features with survival outcomes, but does not provide specific details on the methods or statistics used. More information is needed on the patient cohort, imaging techniques, feature extraction process, and statistical analyses performed.The abstract concludes the findings strongly support further investigation, but does not mention the limitations of the small sample size and retrospective study design. Caution is warranted when drawing conclusions from a preliminary study.The introduction section provides a good overview of the background and motivation for using radiomic analysis of hyperpolarized MRI in prostate cancer. However, it lacks references to specific prior studies applying radiomic methods to hyperpolarized imaging. More background is needed on common radiomic feature extraction techniques and their potential benefits over simple kPL maps for probing cancer metabolism. While the methods describe the process of feature extraction using PyRadiomics software, details are lacking on the specific parameters, settings, and feature classes used. This reduces reproducibility.No description is provided of the methods for survival analysis, including details on the Cox models and multivariate feature selection process.More information should be given on the criteria and definitions used for progression and endpoints like PFS and OS. Ambiguity in endpoint assessment can bias results. The results section focus on findings from the univariate kPL analysis. Details on the multivariate model features and performance are limited, despite this being a key aim.Statistical quantities like p-values and C-indices are presented without accompanying hazard ratios or confidence intervals. Adding these would improve interpretation. The correlation matrix in Figure 3 is described narratively, but the underlying correlation coefficients and significance are not reported. The Discussion does too much interpretation and speculation about the potential biological meaning of the findings given the small sample size. Caution is required when making inferences from a hypothesis-generating study. Limitations around the study design, endpoints, and potential biases are mentioned but should be expanded on. Retrospective studies can suffer from confounding factors. The Discussion does not critically appraise whether the radiomic method provided added value over simply analyzing kPL maps, a key question.
Reviewer 3 Report
Comments and Suggestions for Authors
This paper presents a novel multivariate radiomic framework using hyperpolarized 13C-pyruvate MRI to extract metabolic imaging features in advanced prostate cancer, showing promising correlations with survival outcomes.
My comments:
Introduction:
It would be helpful to more clearly state why radiomic methods, rather than AI or deep learning approaches, were chosen. What are the comparative advantages in this specific context?
Materials and Methods:
Although the manuscript includes a reference to IRB approval and informed consent in the final “Institutional Review Board Statement” section, there is no mention in the main Methods section that participants provided written informed consent. For clarity and transparency—especially in clinical imaging studies involving investigational agents—this information should also appear explicitly in the Methods narrative.
Who performed the segmentations? Were the ROIs manually or semi-automatically delineated? Was interobserver agreement assessed? Given the high dimensionality of extracted features, reproducibility in ROI definition is critical and should be addressed.
Were any efforts made to harmonize image acquisition parameters across subjects, especially given the use of multiple target lesion types (bone, lymph node, liver, primary tumor)?
The approach using C-index-based ranking and hierarchical clustering is interesting. Could the authors comment on whether they tested other dimensionality reduction techniques (LASSO, PCA), and why this method was ultimately chosen?
Discussion:
The authors acknowledge the small sample size as a limitation. Could they discuss in more detail how this impacts model generalizability and what sample size would be needed for external validation?
Given the large number of features and relatively few events, there is a risk of false discovery. A clearer explanation of how the multivariate framework avoids overfitting would improve reader confidence in the conclusions.
Round 2
Reviewer 3 Report
Comments and Suggestions for Authors
This resubmission addresses all the substantive issues that I identified before. The paper is ready for publication.